# Influence of the NiFe/Cu/NiFe Structure Dimensions and Position in External Magnetic Field on Resistance Changes under the Magnetoresistance Effect

**DOI:** 10.3390/ma16134810

**Published:** 2023-07-04

**Authors:** Jakub Kisała, Andrzej Kociubiński, Elżbieta Jartych

**Affiliations:** Department of Electronics and Information Technology, Faculty of Electrical Engineering and Computer Science, Lublin University of Technology, 20-618 Lublin, Poland; j.kisala@pollub.pl (J.K.); a.kociubinski@pollub.pl (A.K.)

**Keywords:** magnetron sputtering, thin films, static magnetic field, giant magnetoresistance

## Abstract

The subject of this work is NiFe/Cu/NiFe thin-film structures made by magnetron sputtering and showing the phenomenon of magnetoresistance. Three series of samples differing in spatial dimensions and thickness of the Cu spacer were produced. During the sputtering process, an external magnetic field of approx. 10 mT was applied to the substrate. Measurements of the resistance of the structures were carried out in the field of neodymium magnets in three different positions of the sample in relation to the direction of the field. The measurements allowed us to indicate in which position the structures of different series achieved the greatest changes in resistance. For each of the three series of layer systems, the nature of changes in the determined coefficient of giant magnetoresistance Δ*R*/*R* remained similar, while for the series with the smallest copper thickness (2.5 nm), the coefficient reached the highest value of about 2.7‰. In addition, impedance measurements were made for the structures of each series in the frequency range from 100 Hz to 100 kHz. For series with a thinner copper layer, a decrease in impedance values was observed in the 10–100 kHz range.

## 1. Introduction

Giant magnetoresistance (GMR) is a phenomenon that has had a significant impact on the development of spintronics, as well as applied technology. The discovery of the phenomenon allowed, among other things, the storage density of hard disks to be increased by constructing smaller heads for reading data. Structures showing the GMR phenomenon are an alternative to Hall sensors, and are widely used in the automotive industry, non-destructive diagnostic systems and medical engineering [1,2,3,4,5,6]. The effect is observed in structures consisting of alternating ferromagnetic layers separated by layers not showing strong magnetic properties. The phenomenon of giant magnetoresistance is based on the scattering of conduction electrons in ferromagnetic layers depending on the compatibility of their spin with the direction of the ferromagnetic layers magnetisation [7,8,9]. A sufficiently strong external magnetic field remagnetises the ferromagnetic layers whose domains tend to orient themselves in the direction of the applied field. This results in a reduction in the scattering of conduction electrons with spin aligned with the direction of the applied magnetic field, while electrons with an opposite spin are scattered much more often. Finally, the electrical resistance of such a structure decreases because, according to Mott’s model, the transport of charges can be treated as two-channel, where electrons with a specific spin flow in each channel [8,10].

GMR structures occur in two main configurations that define the direction of current flow through the sample, i.e., Current In Plane—CIP and Current Perpendicular to Plane—CPP. Depending on the configuration, the structures must be properly designed so that the phenomenon of magnetoresistance can occur in them. In the case of both configurations, it is important that the thickness of the non-magnetic layer is selected to allow the transport of electrons from one ferromagnetic layer to the other without disturbing the path of electrons in the separating layer. Practically, this means that the thickness of the non-magnetic layer should not exceed the mean free path of the electron in the layer material (for copper it is about 20 nm) [11,12]. The phenomenon of giant magnetoresistance depends to the greatest extent on the thickness of the selected layers. In the case of CPP geometry, large values of changes in the giant magnetoresistive effect are observed, especially for multilayer systems and at low temperatures (up to 100% at a temperature of ~20 K [13]). This is mainly due to the fact that in the CPP configuration, electrons must flow through all of the layers of the structure. In the case of the CIP configuration, it is possible to transport electrons within only one layer (magnetic or non-magnetic), and then the giant magnetoresistance phenomenon does not occur or is weakened [14,15]. Therefore, the giant magnetoresistance coefficients usually obtained are much lower (up to 10% at room temperature [16,17]).

The work of the discoverers of the giant magnetoresistance phenomenon presents the characteristics of changes in the resistance of the system depending on the induction of the external magnetic field [18]. The shape of this graph is considered an exemplar of the giant magnetoresistance phenomenon.

The GMR coefficient (Δ*R*/*R*), as one of the main parameters determining giant magnetoresistance, is the ratio of the difference between the resistance of the structure in a specific magnetic field (*R_H_*) and outside the magnetic field related to the resistance of this sample in the zero field (*R*_0_), expressed in percentage:Δ*R*/*R* = (*R_H_* − *R*_0_)/*R*_0._(1)

Studies of thin-film structures with variable spatial parameters allow us to determine the influence of geometry on the obtained resistance changes. Due to a number of variable parameters affecting magnetoresistance, the results of measurements made by different teams of researchers may differ, despite maintaining the same spatial dimensions of the samples, their configuration (CIP or CPP) and location relative to the external magnetic field [16,19,20]. Even the roughness of the substrate can dramatically affect the properties of the structure with a layer thickness of a few nanometres.

Studies on the influence of a number of parameters on changes in the coefficient of giant magnetoresistance have been carried out for some time, leading to many of the applications already mentioned. However, various systems are being used to obtain thin films that differ in elusive details that can have a considerable impact on the structures obtained. Estimating the influence of the parameters under study using the relatively simple magnetron sputtering method could lead to the development of a more complex measuring device in the future. Current research entails determining the technological capabilities of the magnetron sputtering system used.

In this work, FM/NM/FM trilayers in CIP configuration were produced by magnetron sputtering, where FM represents ferromagnetic material, and NM non-magnetic material. The purpose of the study was to evaluate the influence of the thickness of the non-magnetic layer on the resistance changes of the FM/NM/FM system under the giant magnetoresistance phenomenon. Trilayers resistance measurements were also carried out at different positions of the sample relative to the direction of the magnetic field in order to identify the most favourable position for which changes in the Δ*R*/*R* ratio will be the greatest.

## 2. Materials and Methods

### 2.1. Metallic Materials

The materials necessary to make the basic GMR structure should be characterised by high conductivity (around 10^7^ S/m). Ni81Fe19 permalloy with a purity of 99.95% was used as the ferromagnetic material. Permalloys are widely used in magnetic applications due to their high values of relative magnetic permeability (up to 10^5^) and small coercive field (~1 Oe), but their exact values depend on the thickness of the thin layer [21,22,23]. The thickness of the applied ferromagnetic layer has less influence on the phenomenon of giant magnetoresistance than the thickness of the separating layer, but still remains a key factor.

The non-magnetic material separating the ferromagnetic layers in the fabricated structures is copper. Among non-magnetic metals, it is characterised by high electrical conductivity (5.96 × 10^7^ S/m at 20 °C) [24]. This material is also relatively cheap and very widespread. The purity of the copper source was 99.99%.

The non-magnetic layer plays an important role of the magnetic separation of the layers undergoing remagnetisation. Its thickness has a direct bearing on the distance between the ferromagnetic layers. It should be thin enough to allow antiferromagnetic coupling to occur, but not too thin so that the remagnetisation of the layers could take place in a sufficiently low magnetic field. The main purpose of the layer is to magnetically isolate subsequent layers and not disturb the path of electrons [10].

### 2.2. Magnetron Sputtering

The NANO 36™ Kurt J. Lesker^®^ magnetron sputtering system was used to produce the multilayer systems (software v1.0). Magnetron sputtering is a type of ion sputtering distinguished by the use of an additional magnetic field in the form of permanent magnets or electromagnets under the target. The magnetic field, with an induction of about 0.1 T, occurring at the source of the material, traps the electrons above the target surface, leading to the intensification of the process of knocking out the atoms of the desired material from the source. Metallic materials, which are sufficient to produce a structure exhibiting the phenomenon of giant magnetoresistance, require only a DC power supply to the magnetron.

The structures were deposited on a chemically cleaned glass substrate. Kapton tape was used as a technological mask, allowing for easy determination of the shape of the sample. Permalloy and copper layers were produced by subsequent sputtering processes. The parameters of the sputtering processes for the NiFe/Cu/NiFe series 1 system are given in Table 1.

During the process of sputtering ferromagnetic layers along the longer edge of the structure, a magnetic field in the form of neodymium magnets with an induction value of approx. 10 mT was applied. This was used to induce an easy axis of magnetisation in the ferromagnetic layers [9,25]. The value of 10 mT is the result of the use of cylindrical neodymium magnets. In the literature, different values of the applied field can be found from 20 Oe to 200 Oe (which corresponds to the interval 2–20 mT) [9,26]. The thickness of the permalloy layers was 30 nm in each series of samples. The obtained structures haddifferent copper layer thicknesses: 5 nm and 2.5 nm. The thickness of the sputtered layer was monitored during the sputtering process using a quartz resonator. There was also a difference in spatial dimensions between the different series of samples; 3 mm × 10 mm and 2 mm × 20 mm, respectively (see Table 2).

The choice of copper thickness is dictated by the possibilities of obtaining a uniform layer via magnetron sputtering, not exceeding the length of the average free path of the electron and the arbitrary choice of two various thicknesses significantly different from each other. The chosen spatial dimensions are due to the type of technological mask used (Kapton tape), as well as the used measuring stations allowing the measurement of structures of such dimensions. After the technological processes were completed, the mask was removed. The final structures were mounted with graphite paste to the wires in order to carry out resistance measurements. A sample structure can be seen in Figure 1.

### 2.3. Measurement Station

The structures were subjected to DC resistance measurements under the influence of an external magnetic field. All measurements were carried out at room temperature. Strong neodymium magnets mounted on linear guides were used as the source of the external magnetic field. The variable distance between the magnets resulted in a change in the induction of the magnetic field applied to the layer system located between the magnets. The range of changes in the magnetic field induction was 0–0.3 T. Each of the samples were placed in the magnetic field in three different positions relative to the direction of the magnetic field as shown in Figure 2, and in the remainder of the article are referred to as position (a), (b) and (c). For position (a), both the direction of current flow through the structure and plane of the layers are in alignment with the direction of the magnetic field. Position (b) is characterized by the perpendicularity of the direction of current flow and parallelism of the plane of the layers to the direction of the magnetic field. Position (c) from position (b) is differentiated by the perpendicularity of the plane of the sample surface to the external magnetic field.

Measurements of the DC electrical resistance of the trilayers were made using the KeySight 34410A multimeter, (software v1.1.0.0). The LabVIEW environment was used to carry out the measurements. The intensity of measuring current was 100 µA.

In addition, for position (a) of the structures of all three series, impedance measurements were carried out in the frequency range of 100 Hz–100 kHz. These measurements were made using a RuoShui 4090C LCR multimeter. The MATLAB environment was used to carry out the measurements. They consisted of placing the sample outside the magnetic field of a strong neodymium, carrying out a series of measurements with increasing measurement frequency and then repeating the series of measurements when the sample was in a strong magnetic field of 0.5 T.

## 3. Results and Discussion

Figure 3, Figure 4 and Figure 5 present the determined GMR coefficients (Δ*R*/*R*) for three series of NiFe/Cu/NiFe samples as a function of variable magnetic field induction (Δ*R*/*R* = f(*B*)) for three different positions. The measurement points in Figure 3, Figure 4 and Figure 5 represent the result of the arithmetic mean of the coefficients of five different samples of one series.

The average resistance of the structures for the following series was: 820 Ω (series 1), 2465 Ω (series 2) and 2556 Ω (series 3). These values vary due to the series, which is a result of the different geometric dimensions of each series. For all samples, with the increase in the value of the field induction, a decrease in the DC resistance of the sample is noticeable, and thus also an increase in the absolute value of the Δ*R*/*R* coefficient is shown in the figures. Its values are negative, which indicates a decrease in the resistance of the structure under the influence of the magnetic field.

In the case of the (a) and (b) positions, for each of the series of samples, the shape of changes in the GMR coefficient was similar and resembled changes obtained in other tests [27]. The greatest resistance changes, as well as an increase in the GMR coefficient value, were observed in the range of 0–0.1 T. For all series, in the range of induction values of 0.1–0.15 T, a flattening of the characteristics of changes in the GMR factor was noticeable. Further increasing the magnetic field induction resulted in much smaller changes. The smallest resistance changes were observed for the smallest dimensions of the samples from series 1 (10 mm × 3 mm), which had the lowest resistance. For samples with different spatial dimensions, but with the same thickness of the Cu layer (Figure 3 and Figure 4), the coefficient of giant magnetoresistance reached comparable values. For each of the layer systems, the highest GMR factor was observed for position (b) and was about 2.4‰, while for the structure with a thinner copper thickness it was slightly higher, i.e., 2.7‰ (Figure 5). The smallest resistance changes were observed for position (c). The expected magnetization directions of the ferromagnetic layers for each position in the external magnetic field are shown in Figure 6.

Figure 7, Figure 8 and Figure 9 show the frequency dependencies of impedance of structures of each series in position (a). Measurements were made only for position (a), which was due to the capabilities of the bench used. They were intended to observe the change in impedance depending on the frequency of the measurement voltage, and not just the value of the GMR obtained.

The induction value of the strong neodymium magnets used is 0.5 T. For each of the series, a decrease in the impedance of the structure under the influence of an external magnetic field is observed. The slight fluctuations in the impedance of each structure for a zero field and a 0.5 T field seem to coincide. For the structure with the smallest copper thickness of 2.5 nm, there is a noticeable drop in measured impedance between 10 kHz and 100 kHz. The nature of these changes was noticeable for each NiFe(30)/Cu(2.5)/NiFe(30) structures.

## 4. Conclusions

NiFe/Cu/NiFe thin-film structures were produced by magnetron sputtering. Measurements of the electrical resistance of three series of thin-film resistors were carried out. The samples differed in spatial dimensions (width and length) and the thickness of the copper layer. The conducted tests indicate that the direction of the current flow relative to the direction of the magnetic field influences the value of the observed magnetoresistance changes in the produced structures (position (a) and (b)). Additionally, the position of the sample plane relative to the direction of the magnetic field affects the value of Δ*R*/*R* (position (b) and (c)). Position (c) turned out to be the least sensitive to changes in the value of the magnetic field. In the case of positions (a) and (b), changes in resistance were visible for each of the structures, suggesting that the phenomenon of giant magnetoresistance was observed. This is also evidenced by the shape of the obtained Δ*R*/*R* =f(*B*) characteristics. However, position (a) does not seem to be a natural position for an element that could be used as a potential magnetic field sensor. It can therefore be concluded that the position of the structure relative to the direction of the magnetic field induction is essential. In the future, we plan to study the dependence of changes in resistance for a larger number of different positions of the sample in an external magnetic field in order to determine the possibility of using this type of structures made by sputtering method as sensors operating in two planes.

The application of an external magnetic field during the sputtering process on the target is to induce an easy axis of magnetization in the ferromagnetic layers. The lines of this field are in line with the longer edge of the planes of the resulting layers. In the case of position (c), the observed resistance changes in the range of 0–0.3 T are the smallest because the domains become magnetized perpendicular to the plane of the thin layer of ferromagnetic material. For position (a), assuming the initial antiparallel configuration of the ferromagnetic layers, one of the ferromagnetic layers is remagnetised by a resultant angle of 180°. For position (b), both layers become magnetised by 90°.

The largest resistance changes were observed in the case of the NiFe(30)/Cu(2.5)/NiFe(30) system, and the smallest for the structure with a copper thickness of 5 nm and sample dimensions of 3 mm × 10 mm, also characterised by the lowest electrical resistance. This fact allows us to conclude that the dimensions of the produced thin-film system, as well as the thickness of the copper, affect the value of the observed resistance changes within the giant magnetoresistance phenomenon. For structures of the NiFe(30)/Cu(5)/NiFe(30) configuration, the coefficient of giant magnetoresistance reached values up to 2.4‰. For the series with a copper thickness of 2.5 nm, an increase in the GMR factor to about 2.7‰ was noticeable. The obtained values of the coefficient of giant magnetoresistance were small compared to values reported in other studies. In the future, we plan to make more complex structures and obtain higher GMR values.

Additional measurements made with LCR multimeter on the structures of each series in position (a) allowed us to observe the frequency characteristics of impedance. A drop in the impedance values of the structures in a strong magnetic field were observed. However, undesirable minor fluctuations are noticeable, both for field induction of 0 T and 0.5 T, depending on the frequency of the measurement voltage. These fluctuations were very similar for out-of-field and in-field measurements. For structures with the smallest copper thickness, 2.5 nm, a decrease in impedance was observed in the frequency range above 10 kHz. However, the difference in impedance between the two measured states is constant. This result indicates the appearance of parasitic capacitance in the fabricated structure related to the smaller relative distance of the ferromagnetic layers.

## Figures and Tables

**Figure 1 materials-16-04810-f001:**
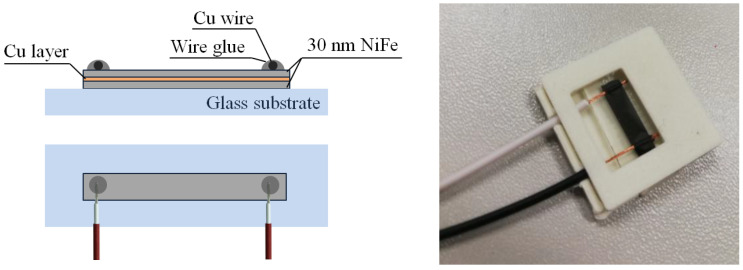
Final structure mounted to wires by graphite glue; schematic side and top view and photo of real sample.

**Figure 2 materials-16-04810-f002:**
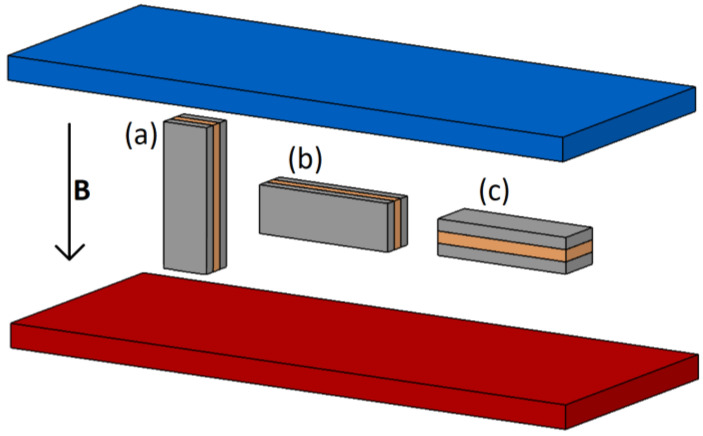
Three positions of samples in external magnetic field; (a), (b) and (c), where *B* is magnetic induction of two neodymium magnets (blue and red).

**Figure 3 materials-16-04810-f003:**
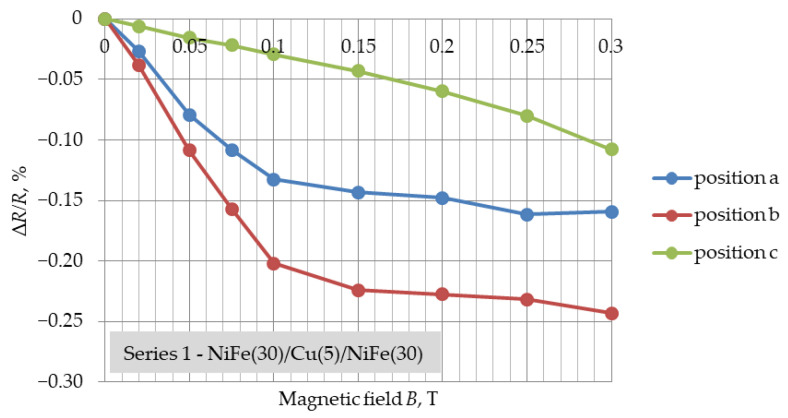
Results of Δ*R*/*R* = f(*B*) for three positions of NiFe(30)/Cu(5)/NiFe(30), dimensions: 10 mm × 3 mm, magnetic field range: 0–0.3 T.

**Figure 4 materials-16-04810-f004:**
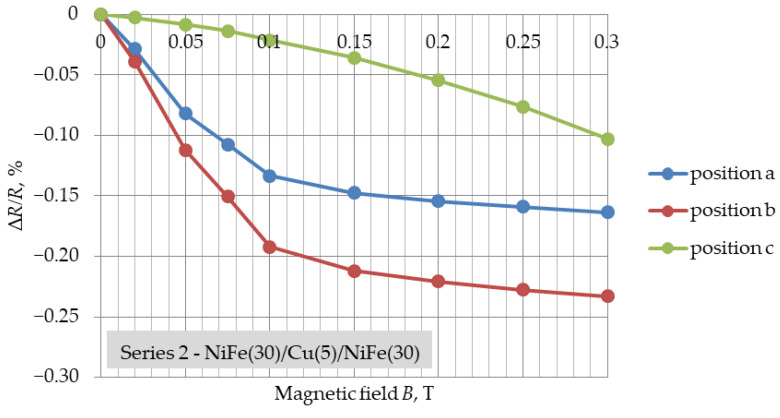
Results of Δ*R*/*R* = f(*B*) for three positions of NiFe(30)/Cu(5)/NiFe(30), dimensions: 20 mm × 2 mm, magnetic field range: 0–0.3 T.

**Figure 5 materials-16-04810-f005:**
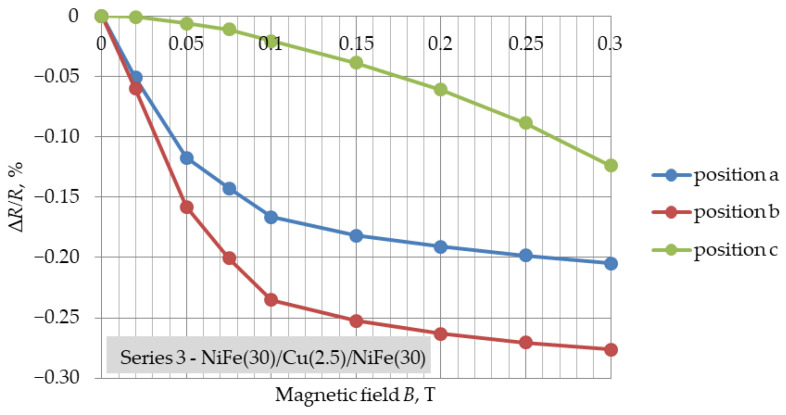
Results of Δ*R*/*R* = f(*B*) for three positions of NiFe(30)/Cu(2.5)/NiFe(30), dimensions: 20 mm × 2 mm, magnetic field range: 0–0.3 T.

**Figure 6 materials-16-04810-f006:**
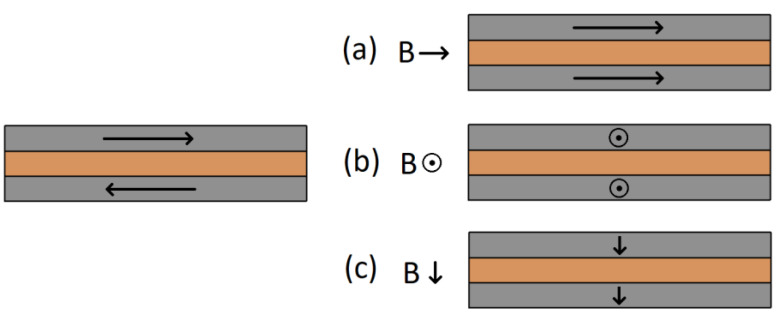
Directions of magnetization of ferromagnetic layers: left—in the absence of magnetic field, right—in the external magnetic field; for each position (a), (b) and (c) the directions of magnetization are parallel to the induction *B* of the external magnetic field.

**Figure 7 materials-16-04810-f007:**
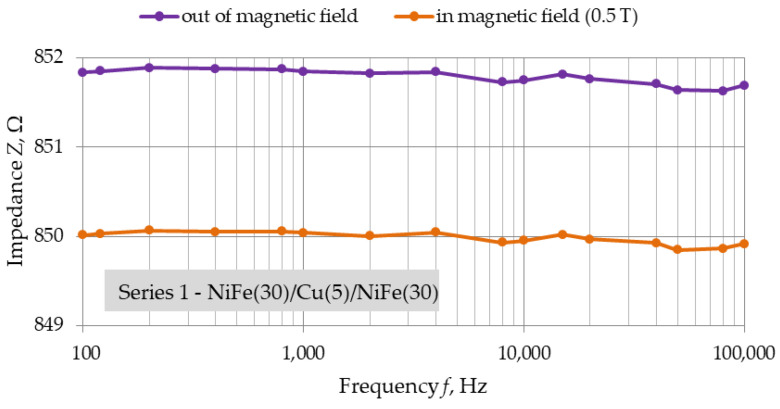
Position (a), *Z* = f(*f*), Series 1—NiFe(30)/Cu(5)/NiFe(30) structure for alternating voltage, frequency range: 100 Hz–100 kHz, dimensions: 10 mm × 3 mm.

**Figure 8 materials-16-04810-f008:**
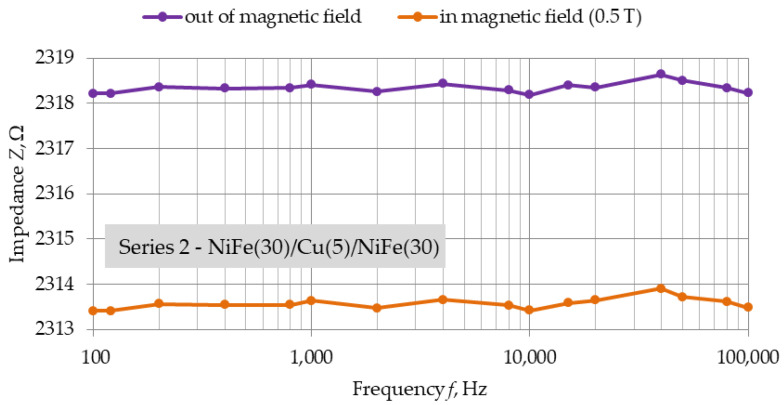
Position (a), *Z* = f(*f*), Series 2—NiFe(30)/Cu(5)/NiFe(30) structure for alternating voltage, frequency range: 100 Hz–100 kHz, dimensions: 20 mm × 2 mm.

**Figure 9 materials-16-04810-f009:**
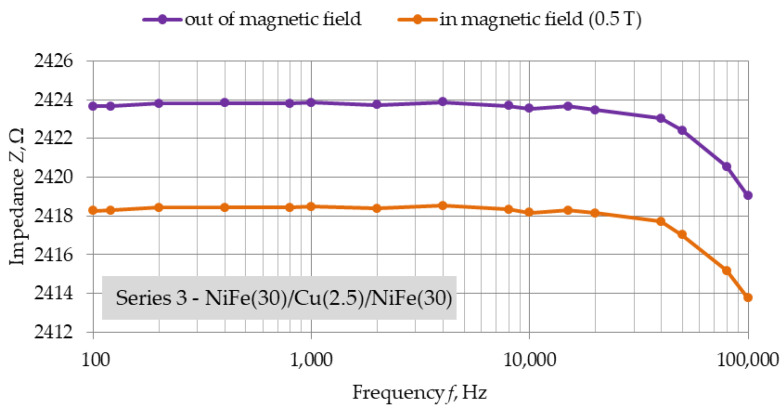
Position (a), *Z* = f(*f*), Series 3—NiFe(30)/Cu(2.5)/NiFe(30) structure for alternating voltage, frequency range: 100 Hz–100 kHz, dimensions: 20 mm × 2 mm.

**Table 1 materials-16-04810-t001:** Parameters of magnetron sputtering processes for series 1.

Pressure [Torr]	Material	Thickness [nm]	Plasma Power Density [W/cm^−2^]	Argon Flow Rate [sccm]	Approximate Deposition Rate [Å/s]	Deposition Time [min]
10^−7^	Ni81Fe19	30	90	85	0.33	15
	Cu	5	90	85	0.24	3.5
	Ni81Fe19	30	65	85	0.28	18

**Table 2 materials-16-04810-t002:** Parameters of structure dimensions of three series.

Series	Permalloy Thickness [nm]	Copper Thickness [nm]	Length [mm]	Width [mm]
1	30	5	10	3
2	30	5	20	2
3	30	2.5	20	2

## Data Availability

Not applicable.

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
