# Peer review of "Influence of the NiFe/Cu/NiFe Structure Dimensions and Position in External Magnetic Field on Resistance Changes under the Magnetoresistance Effect"

_materials, 2023, doi:10.3390/ma16134810_

Round 1

Reviewer 1 Report (Previous Reviewer 1)

All the comments are addressed properly.

Minor editing requires. Overall good.

Reviewer 2 Report (Previous Reviewer 2)

The authors have solved most of my doubts and improved the quality of the paper. However, I still insist on my opinion that the novelty and originality of this paper is not enough. 

Minor editing of English language is required.

Reviewer 3 Report (Previous Reviewer 3)

No more comments.

No more comments.

This manuscript is a resubmission of an earlier submission. The following is a list of the peer review reports and author responses from that submission.

Round 1

Reviewer 1 Report

This paper has studied magnetoresistance with NiFe/Cu/NiFe thin film structures that was made by magnetron sputtering. Samples with different spatial dimensions were prepared and measurements were done with different orientations. Although author provide some positions and conditions where maximum resistance change is observed, some points need to be addressed before publishing.

1)    It was mentioned in the paper during the sputtering of the ferromagnetic layer 10mT magnetic field was applied. What happened if the deposition is done at different magnetic field?

2)    Three different dimension was chosen. Is there any reason for choosing those dimensions in particular?

3)    Show the actual picture of the device with the graphite connection and all. That will help audience understand the system better.

4)    Three different position of the device is not very clear. Need better description. Figure 1 is hard to follow. Need labeling of the layers. Perhaps a better picture would be helpful.

5)    Need better explanation of why the resistance change between a, b, c. From the diagram it looks like magnetic field direction is perpendicular to the ferromagnetic film. Since the easy axis is also perpendicular, should not this orientation give minimum resistance?

6)    What happened to the resistance vs Frequency plot with position b and c?

7)    Need a better description of why this study is so important, what is new about this study and a better explanation? Also, how this study is useful for future magnetoresistance devices.

8)    Minor: There are certain grammatical/typographical errors. The manuscript will benefit from a thorough proof-reading

There are certain grammatical/typographical errors. The manuscript will benefit from a thorough proof-reading

Reviewer 2 Report

The influences of sample spatial dimension and its position relative to the magnetic field on GMR coefficient were studied in this paper. However, the research content is not important and meaningful enough and some experimental results are easy to predict. For me, this article does not meet the requirement of the journal.

1. The related research on GMR effect has been carried out for many years and as mentioned in the manuscript (Ref [16, 19, 20, 27]), the influences of dimension parameter and magnetic field direction on the GMR coefficient were also previously studied, so what’s the novelty or innovation of this paper?

2. The “Results and Discussion” part only briefly describes the experimental results and there is no any physical explanation for the experimental phenomenon.

3. As mentioned in line 177-178, the smallest resistance changes were observed for the smallest dimensions of the samples from series 1, i.e., 10 mm × 3 mm. This description is incorrect from Fig. 2 and 3. The resistance changes at position b of series 1 in Fig. 2 are even larger than that of series 2 in Fig. 3. In addition, the curves for position c in Fig. 2 and 3 coincide perfectly, which is difficult to understand and makes me to doubt the correctness of the measured data.

4. The material thickness can not be matched with the deposition rate and deposition time (miswritten as deposition rate) in Table 1.

5. There are several grammar errors in the language writing, for example, the word “exemplary” should be replaced with its noun form.

The English language of this article is moderate and still needs to be modified.

Reviewer 3 Report

   This manuscript studies the Influence of the NiFe/Cu/NiFe structure dimensions and position in external magnetic field on GMR changes, and I have the following comments:

(1)      The most drawback of this manuscript is the lack of physical explanation and analysis on most of experimental results, which need significant improvements.

(2)      For the samples in the external magnetic field a, b and c, it’s not clear why the position c obtains the largest GMR effect since the demagnetizing field should be the smallest for the position a?

(3)      For the results 2-4, more analysis based on physical model or micromagnetic simulation should be carried on to explain the effect of separation layers, which should have been explored by other researchers years ago.

(4)      The effect of excitation frequency should be explained, which may be related to the relaxation of magnetic domain at the high frequency.

(5)      The reported maximum GMR ratio is only 0.27%, which is far below the previous reports.

The Quality of English Language is acceptable.